

# Ice flow velocity as a sensitive indicator of glacier state

Martin Stocker-Waldhuber[1/2], Andrea Fischer[1], Kay Helfricht[1], Michael Kuhn[3]

[1]Institute for Interdisciplinary Mountain Research, Austrian Academy of Sciences, Innsbruck, 6020, Austria
[2]Department of Geography, Physical Geography, Catholic University of Eichstätt-Ingolstadt, 85072, Germany
[3]Institute of Atmospheric and Cryospheric Sciences, University of Innsbruck, Innsbruck, 6020, Austria

*Correspondence to*: Martin Stocker-Waldhuber (martin.stocker-waldhuber@oeaw.ac.at)

**Abstract.** Climatic forcing affects glacier length changes, mass balance and ice flow dynamics on different time scales and also dependent on topography. The first two of these parameters are operationally used for glacier monitoring, whereas only a few time series of glacier dynamics exist with the potential to serve as long-term indicators of glacier response to climate change. With more than 100 years of measurements of ice flow velocities at stakes and stone lines on Hintereisferner (HEF)

and more than 50 years on Kesselwandferner (KWF), records of annual velocity change are as long as records of glacier fluctuations. Interannual variations of ice flow velocities and shorter supporting interpretations of long-term records have been measured on Gepatschferner (GPF) and Taschachferner (TSF) for nearly 10 years. The ice flow velocities on Hintereisferner and especially on Kesselwandferner show great variations between advancing and retreating periods, with magnitudes increasing from the highest to the lowest stakes, making ice flow records at ablation stakes a very sensitive

indicator of glacier state. Since the end of the latest glacier advances from the 1970s to the 1980s, the ice flow velocities have decreased continuously, a strong sign of the severe retreat of the glaciers in recent decades.

## 1 Introduction

The fluctuation of glaciers has become an icon of climate change, after Penck and Brückner (1909) established the theory of ice ages, later confirmed by isotope analysis on polar ice cores and theoretically explained by Milankovitch. First monitoring

efforts focused on recording the changing positions of glacier termini, which started in the 17[th] century and were systematically organized in the late 19[th] century. In the case of catastrophic glacier advances, as reported several times during the Little Ice Age, for instance, for Vernagtferner in the Ötztal Alps (Nicolussi, 2012), local observers often reported the velocity of terminus advances over short periods. At that stage of development, glaciological theory and monitoring techniques, the monitoring of horizontal ice flow velocities was already well established for Alpine glaciers, with 5 glaciers

with stone line velocity records among 20 glaciers regularly monitored for length changes in the Eastern Alps, for example, at the glaciers of Pasterze (Nicolussi and Patzelt, 2001), Vernagtferner (Braun et al. 2012) or Hintereisferner (Span et al., 1997), or in the Western Alps at Rhone glacier (Mercanton, 1916; Roethlisberger, 1963) or Mer de Glace (Berthiere and Vincent, 2012). For the large outlet glaciers of Greenland and Antarctica, ice flow velocity records have been derived from





satellite radar interferometry (e.g. Rignot and Kanagaratnam, 2006; Moon et al., 2012, Hogg et al., 2017, Rott et al., 2017) and show that changing patterns of ice flow velocity play an important role in the reaction of polar glacier systems to climate change. Feature Tracking and DinSAR both do not reveal full 3D velocity information and need additional assumptions. These data are thus valuable for change mapping, but need to be complemented by detailed process studies to understand

uncertainties and limitations, the later also emerging from the presence of stable features.

For Alpine glaciers, monitoring of velocity records received less attention after the turbulent decades of the First and Second World War. Development of glaciological programmes focused on hydrological programmes and mass balance programmes, as the problem of glacier flow was solved by deformation measurements on Hintereisferner and in the theoretical work of Finsterwalder (1907) and Hess (1924). Nowadays, when estimates of the global glaciers' contribution to sea level rise is one

of the urgent topics of research (e.g. Jacob et al., 2013), and estimates of the state of large glacier ensembles are needed, glacier flow velocities, as potentially remote-sensing based glacier parameters, should be revisited for their suitability of replacing parameters based on mass balance theory. For example ELA, which played a major role in large-scale data collections on global climate change, has been observed to be above summits and thus undefined for Eastern Alpine glaciers for much of the last decade (WGMS, 2017).

In this paper, four long time series of ice flow velocities are revisited and compared with classical in situ mass balance parameters (Hoinkes, 1970) and ALS data (Abermann et al., 2010) to find out if large-scale ice flow velocity monitoring would be a potential alternative to mass balance parameters for regional glacier monitoring on an annual basis. Flow velocity records are measured in situ at stone lines (flow path) or at stakes (3D velocities), with stakes annually set back to their original position. Long-term velocity data are recorded at annual resolution, shorter time series also reveal seasonal

variabilities. All of the investigated glaciers are also subject to long-term measurements of glacier fluctuations (Groß, 2018); area and volume change have been recorded for four time steps from LIA maximum onwards (Fischer et al., 2015).

Without further discussion of the potential and limitations of different remote-sensing sensors and techniques, this paper focuses on presenting empirical ice flow velocity records on well investigated mountain glaciers and their relation to other in situ monitoring parameters.

## 2 Glacier sites and data

The ice flow velocities have been recorded on four of the largest glaciers in the Ötztal Alps. Hintereisferner (HEF), Kesselwandferner (KWF), Gepatschferner (GPF) and Taschachferner (TSF) are neighbouring glaciers (Figure 1) but differ in size, aspect and elevation ranges (Table 1).

HEF, a typical valley glacier, has a long tradition of hydrological, meteorological, geophysical and glaciological

investigations (e.g. Blümcke und Hess, 1899; Förtsch and Vidal, 1956; Hoinkes and Steinacker, 1975; Kuhn et al., 1999; Fischer, 2010; Helfricht et al., 2014; Strasser et al., 2018). Ice flow velocities on HEF have been sporadically measured at ablation stakes but almost annually for more than 100 years at stone lines (Span et al., 1997). Line 3 on HEF is the oldest



stone line, started in 1895 and lasting until 1985 when the glacier retreated behind the location of the profile. Records at Line 6 were started in 1932/33 and at Line 7 in 2013/14. In situ mass balances have been measured since 1952 (Hoinkes, 1970; Fischer, 2010; Fischer et al. 2013; Strasser et al., 2018).

The investigations on KWF are historically linked to those on HEF (Kuhn et al., 1985) with the same long-term

investigations of length variations since 1884 and mass balances since 1952 (Fischer et al., 2014). The terminus of KWF detached from the tongue of HEF in 1914. Velocity measurements at this plateau glacier were started in 1965 by Schneider (1970) at ablation and accumulation stakes along the centre flow line of the glacier. A comparison between direct and geodetic measurements on KWF as well as on HEF was made by Fischer (2011).

GPF is the second largest glacier of the Austrian Alps (Fischer et al., 2015). The main glacier rests on a wide but hilly

plateau and the tongue descends through a narrow valley. After early first mappings (Sonklar, 1860; Finsterwalder, 1928), GPF was subject to geophysical investigations (Giese, 1963), photogrammetric analyses (Keutterling and Thomas, 2006) and is one of the Eastern Alpine key research sites, with extensive knowledge on its Holocene fluctuations (Nicolussi and Patzelt, 2001). Recently, Gepatschferner became part of a detailed study on geomorphodynamics by Heckmann et al. (2012). In this study, the stake network at the glacier tongue was extended from three stakes, where velocities have been measured since

2009, to 16 stakes in 2012. The stake velocity records on TSF were started together with those on GPF in 2009 at three positions.

On both glaciers, GPF and TSF, the position of the stakes is measured several times during summer months, allowing a discussion of the seasonal variability. In contrast, the annual velocity records at HEF and KWF can be discussed in relation to the long-term monitoring of glacier mass balances.

**3 Methods**

Based on the historical development of geodetic techniques, different methods came into operation on these glaciers during the past century. Trigonometric networks were installed in 1894 on HEF (Blümcke and Hess, 1899) and in 1965/66 on KWF (Schneider, 1970) to determine glacier surface velocities with a theodolite at stone lines on HEF and ablation stakes on both glaciers (Figure 1). On HEF and KWF, stake velocities were measured using a theodolite and tachymeter until 2009, when

DGPS measurements came into operation. On GPF and TSF, the full series was measured by DGPS.

The velocity records are compared to direct and geodetic mass balance measurements from Hoinkes (1970), Schneider (1970), Fischer (2010), Fischer et al. (2013), Stocker-Waldhuber et al. (2017) and Strasser et al. (2018) on HEF, KWF and GPF. In these publications, the surface mass balances were derived from stakes and snow pits by using the direct glaciological method (Hoinkes, 1970). Additionally, DEMs (Digital Elevation Models) and DODs (DEMs of Difference)

from photogrammetric or high-resolution ALS (Airborne Lase Scan) data came into operation to determine volume and elevation changes (Abermann et al. 2010).



### 3.1 Velocity measurements at stone lines

The method of stone lines (Heim, 1885; Hess, 1904) was used only on HEF at three profiles. The position of several stones and their distance to each other is fixed within a defined cross-profile. The number of stones depends on the glacier width and thus varies in time with any expansion or reduction of the glacier. The position of the stones was measured initially with

tachymetric systems and since 2009 with DGPS. The distance between the original defined position of the stone within the profile and the position in the subsequent year is measured using a measuring tape and represents the flow path of the stone within one year. Afterwards the stones are moved to their initial position. From 2009 the flow path was calculated from the measured DGPS positions, but the measuring tape is still used for control. The annual velocities at the stone lines are given as the mean annual values of the stones in the profile.

Earlier data for Line 6 (before 1932/33) and Line 7 (before 2013/14) were complemented by velocity records from ablation stakes for periods when the stake was reinstalled at its original position. The stakes are located at the central flow line of the glacier, thus representing the maximum flow velocity at the profile. A relation of 80% between the mean velocity from the stone line and the maximum velocity at stakes located at the centre of profile (Span and Kuhn, 2003) was taken to compare the stake values with the mean values from the stone lines.

### 3.2 Velocity measurements at stakes

Velocity measurements at ablation and accumulation stakes are used on KWF, GPF, TSF and partly on HEF. The position of the stakes and their motion on KWF is measured at the top of the stake and calculated to its base point. This has the advantage that the measured velocity is not affected by surface changes of accumulation or ablation. Figure 2 by Schneider (1970) shows the components of the velocity vector (d) at the base point of the stake within the accumulation area (left side)

and the ablation area (right side) between two points in time (t1, t2) depending on submergence (negative value of v) and emergence (positive value of v). This definition coincides with the definition of submergence and emergence in Cogley et al., (2011). The vertical motion can be calculated as the remainder of the absolute elevation change of the surface (Δd) and the accumulation or ablation (Δa) or from the elevation change due to the sloping surface (Δh) and the vertical component (Δz) of the velocity vector (d) (Schneider, 1970). The difference between the actual flow path (d), which is the length of the

velocity vector, to the horizontal motion (Δs), which is the projected velocity, results from the vertical component (Δz). Annual values of the horizontal flow velocity (Δs/a) as well as the vertical motion values of submergence and emergence were calculated for 365 days. The horizontal velocity component (s) and the vertical component (v) in Schneider (1970) correlate to the definitions of (u) and (w) in Cuffey and Paterson (2010).

### 4 Accuracies

The uncertainty of the stone line measurements, determined with a measuring tape, can only be estimated, depending on distance, surface roughness and possible slipping of stones. An uncertainty of the stone line velocity of 5% of the annual



distance should be a good estimate. It must be noted that the velocities of the stone lines refer to the flow path (d) of the stones in contrast to the horizontal component of the velocity vector ($\Delta s$), which is given for the stake values of the other glaciers.

The investigations on KWF aim at very high accuracies, which should be in the range of ±5 cm per measurement or at least

±10 cm for the period. The stakes are reinstalled annually at their initial position determined with theodolite, tachymeter or DGPS with RTK (real time kinematic) procedure. Redrillings and measurements are conducted with a rod level for exact perpendicular conditions. The reflector or the DGPS antenna is directly mounted to the top of the stake. Therefore duraluminium stakes with rigid connection are used on KWF as ablation stakes (Ø = 2 cm) and thicker accumulation stakes with a diameter of 5 cm for the necessary resistance against snow pressure.

In contrast to the stakes on KWF, ice flow velocities at GPF and TSF are measured at wooden ablation stakes with DGPS and post-processing procedure, the measured positions refer to the glacier surface. Thus horizontal accuracies on these glaciers are ±10 cm per measurement or ±20 cm for the period at its best. Additionally, shading effects of the surrounding topography at TSF and GPF lead to higher uncertainties for the DGPS measurements. Stake positions are measured several times during summer seasons, allowing a comparison of shorter time scales.

**5 Results**

**5.1 Hintereisferner**

Three stone line records on HEF indicate the variation of glacier surface velocities for different periods, in total for more than 100 years (Figure 3). Three periods with increasing surface velocities were recorded on HEF. The first and most extensive acceleration of surface velocity happened before 1920, with a maximum velocity of more than 120 m per year in

1919 (Hess, 1924). This acceleration caused a small advance of the glacier terminus in subsequent years. The second period was recorded from 1935 to the early 1940s and the most recent one during the 1970s. During that time the surface mass balance of the glacier was positive for several years, which is even more evident in the geodetic results. Since 1980, surface velocities on HEF have continuously decreased at the stone lines to about 4 m per year in the most recent years at Line 6, and to about 7-8 m per year at Line 7. This continuous decrease is accompanied by strong negative mass balances in the

most recent decades.

**5.2 Kesselwandferner**

On KWF measurements were started for the hydrological year 1965/66, including horizontal and vertical flow velocities (Figure 4 and Figure 5). These long-term investigations document different glacier states at a longitudinal profile of up to ten accumulation and ablation stakes. There are two main contrasting periods, the first one from the start of the measurements to

1985, and the second period since then. During the first period the glacier was in an advancing state because of positive mass balances. During that time 75% of the measured glaciers in Austria and Switzerland presented positive length changes,



caused by positive mass balances as a result of decreasing summer temperature and increasing annual precipitation (Patzelt, 1985), probably a consequence of global dimming (Wild et al., 2007; Braun et al., 2012). The surface velocity of the glacier increased, but with decreasing magnitudes from the terminus (L10) to the uppermost stake (L1) within the accumulation area. This means a large velocity gradient along the glacier, with maximum annual velocities of about 90 m per year at the

terminus, in contrast with a few metres per year at the highest elevations. The gradient of the vertical velocities was also big, with a submergence of up to 3 m per year within the accumulation zone to an emergence of up to 5 m per year at the lowermost stake. During that time the ELA (equilibrium line altitude) shifted to lower elevations which can be seen as the transition from submergence to emergence from stake to stake.

The advancing state of the glacier ended in 1985, followed by a sharp decrease of the surface velocities and a reduction of
the velocity gradients. Negative vertical velocities became gradually positive around 1990 at stakes L8 and L7, in 2005 at L6 and in 2016 at L5, representing a shift of the ELA to higher elevations. The latest positive mass balance occurred in 2015, with an immediate response in the horizontal and vertical velocities. At the lowermost stake L10, velocity decreased rapidly to almost nil because of the decreasing mass supply to the terminus. This area became ice-free in 2010.

**5.3 Gepatschferner and Taschachferner**

The time series on GPF and TSF were started in 2009/10 and the stake network on GPF was extended in 2012. During these measurements, annual velocity fluctuations were small, especially at the three stakes on TSF (54, 55 and 56). At the lowermost stake 54, the horizontal velocities were less than 10 m per year during the whole period. The two higher-altitude points 55 and 56 returned velocities of 30 to around 40 m per year. The higher values at 55 compared to 56 are caused by topographic effects, with a steepening of the glacier tongue and a narrowing of the cross section from stake 56 towards stake
55 (Figure 6).

On GPF a general trend of decreasing surface velocities was found at all stakes, with higher rates from the stakes at the upper cross-profile (71-75) to the terminus. At stake 62, a funnel-shaped surface depression, caused by an evacuation of subglacial sediments due to a heavy precipitation event, led to a limited increase of surface velocity in that area and a later decrease to almost nil (Stocker-Waldhuber et. al. 2017). In total, a general slowdown of velocities at the tongue of GPF was
found since the beginning of the measurements.

An example of interannual fluctuations in surface velocity is given in Figure 7 for stake 65 at GPF, which has been measured from 2009 with least data gaps since then. The velocities are given as mean values per day to make the different time periods of the stake readings comparable. During the winter seasons velocities generally decrease. Maximum values were typically found in August each year, except for the years 2013 and 2014 with earlier peaks in July. A comparison with the geodetic
elevation change at the stake shows an expected inverse process. During winter months the elevation change of the surface is close to zero or even positive, while surface velocity is decreasing. The opposite process is found during the summer season, when the highest surface velocities go along with the most negative elevation change.



## 6 Discussion and conclusions

The long-term investigations of the surface velocities at these glaciers document the state of each glacier and its response to a climate signal. Three periods with accelerating velocities caused by positive mass balances were found in the longest time series on HEF. A time shift of the maximum values from higher to lower stone line profiles indicates the response time of

the tongue. Despite the increase in surface velocities during these three periods, mass gain on HEF was insufficient for the terminus to advance, except for a small advance during the 1920s. Therefore responses of velocity fields are more sensitive to climate forcing than terminus fluctuations for this glacier. In contrast to HEF, the terminus of KWF advanced by more than 250 m from the 1970s to the 1980s (Patzelt, 1985; Fischer et al., 2018). KWF presents an immediate response at all profiles concurrently, which means that a mass gain or increase of the thickness within the accumulation area causes an

increase of the emergence at the lowermost stakes within one year.

During glacier retreat the transition from submergence to emergence shifts to higher elevations, as was found on KWF from 1986 to the present. As a consequence, the magnitude of the vertical velocities decreases, which leads to increasing retreat rates of the terminus but at the same time to an increase of the thickness at higher elevations in case of a positive mass balance. Despite the effect of the mass balance, according to the continuity equation, local thickness changes can also be

caused by convergent or divergent glacier flow.

The investigations show that the magnitude of the fluctuations of the surface velocities is higher at the ablation stakes compared to those within the accumulation area and even higher at the lowermost stakes. This means that velocity measurements, especially at ablation stakes, are very well suited for documenting the glacier state, even more so at a fast reacting glacier like KWF. This is supported by a linear regression of annual mean specific balance (b) of the total glacier

area of HEF and KWF versus the mean specific balance of their accumulation areas (bc) for the period 1965/66 – 1999/00 by Span and Kuhn (2003). They found nearly identical correlation coefficients for the two balances, while the standard deviation σ(b) was twice as high as σ(bc), documenting the higher sensitivity of the ablation areas to mass balance changes of the two glaciers.

On GPF and TSF these measurements were conducted exclusively at stakes at the tongue where the decreasing velocity rates

represent the retreating state of the glacier. The decreasing velocities were found especially on GPF, in contrast to TSF, where the velocity rates remained almost constant during these investigations. As both glaciers have a comparable topography, with a large catchment area and a narrow tongue, it may be inferred that TSF is closer to equilibrium than GPF.

The interannual fluctuations on GPF (Figure 7) represent the typical acceleration of the glacier during summer months, which is well known and was, for example, already measured on HEF from 1900 to 1904 by Blümcke and Finsterwalder

(1905). The accuracy of the measurements on GPF varies over time and depends on shading effects of the surrounding topography. The peak in 2013 shows the greatest uncertainty and is thus not representative for the actual surface velocity. The peaks look like jerky movements of the glacier but cannot be clearly identified as such.





For the investigated temperate mountain glaciers, ice flow velocity is a glaciological parameter that reacts very quickly to changes in the forcing. As conventional parameters like ELA tend to be above summit for the investigated glaciers under current conditions and specific mass balance is affected by rapid changes in area, monitoring of ice flow can be recommended as additionally surveyed parameter at mountain glaciers. Modern methods like DGPS and remote sensing

need only limited extra effort, so that this should be feasible and increases the information available on the currently quickly receding mountain glaciers in a transient state.

**Data availability**

Velocity data of the four glaciers Hintereisferner, Kesselwandferner, Gepatscherner and Taschachferner are available on request from the corresponding author and will be accessible via Pangaea (www.pangaea.de).

**Competing interests**

The authors declare that they have no conflict of interest.

**Acknowledgements**

Maintaining long-term monitoring is always a challenging task and requires financial support and the help of numerous people, to whom we would like to gratefully express our thanks. Terminus variations of the glaciers and the velocity records

on HEF relate to the annual measurements of the Austrian Alpine Club. Mass balance terms are provided by the World Glacier Monitoring Service (WGMS) and the Institute of Atmospheric and Cryospheric Sciences (ACINN). Research (project PROSA) on GPF was funded by DFG (SCHM 472/16-1, SCHM 472/17-2 and BE 1118/33-3) and FWF (I 894-N24 and I 1646-N19) and funding continues from glacier ski resort Kaunertaler Gletscher GmbH and Tiroler Wasserkraft AG (TIWAG), which also supports the measurements on TSF. We want to thank H. Schneider who started the velocity records

on KWF and continued them for more than 50 years. These records are now supported by the non-profit organisation Glacier and Climate. We want to thank B. Scott for editing the English.

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





**Table 1: Geographic location and characteristic numbers of GPF, HEF, KWF and TSF from Austrian Glacier Inventory 3 of 2006 (Fischer et al. 2015) and the year of the start of velocity measurements (meas. since). exp: exposition, Sc: accumulation area, Sa: Ablation area.**

| name | location | exp. Sc | exp. Sa | altitude range [m] | area [km²] | meas. since |
|------|----------|---------|---------|--------------------|-----------|-------------|
| GPF | 46.85°N, 10.75°E | NE | N | 2068-3535 | 16.62 | 2009/10 |
| HEF | 46.79'N, 10.75°E | E | NE | 2399-3715 | 7.49 | 1894/95 |
| KWF | 46.84°N, 10.79°E | SE | E | 2698-3496 | 3.82 | 1965/66 |
| TSF | 46.90°N, 10.86°E | N | NW | 2208-3756 | 5.71 | 2009/10 |

**Figure Captions**

Figure 1: Location of the stone lines (3, 6 and 7) on Hintereisferner (HEF) and stakes on Kesselwandferner (KWF), Taschachferner (TSF) and Gepatschferner (GPF). On GPF, stakes 60 to 66 are longitudinal stakes from the glacier snout upwards to the first cross profile 67 to 70 (from the orographic right to the left). Stakes 71 to 75 are located at the root zone of the tongue as a cross profile. The glacier area was taken from the Austrian Glacier Inventories (GI) from LIA (little ice age) around 1850, GI1 from 1969, GI2 from 1998 and GI3 from 2006 (Fischer et al., 2015). Background:

Orthophoto from 2015; data source: Land Tirol – data.tirol.gv.at.

Figure 2: Drawings by Schneider (1970) of the motion of a stake and changes at the glacier surface (Oberfl.) between two time steps (t1, t2) within the accumulation area (left) and the ablation area (right). d: flow path (length of the velocity vector), v: vertical velocity, Δs: horizontal velocity (projected velocity), Δd: absolute surface elevation change, Δa: relative surface elevation change from accumulation or ablation.

Figure 3: The mean annual flow path of the stones at Lines 3, 6 and 7 on HEF since 1894/95 (= 1895). Data series extended since Span et al. (1997) and annual specific surface mass balance (b direct) since 1953 (Strasser et al., 2018; WGMS, 2017; original data: Hess, 1924) as well as the geodetic balances from DoDs (b geodetic) by Fischer (2011). Location of the stone lines s. Figure 1.

Figure 4: Annual horizontal flow velocities (Δs/a) at the accumulation and ablation stakes on KWF (e.g. the year 2015 refers

to the hydrological year 2014/2015) and the specific surface mass balance (b direct) (Strasser et al., 2018; WGMS, 2017) as well as the geodetic balances from DoDs (b geodetic) by Fischer (2011). Location of the stakes s. Figure 1.

Figure 5: Annual vertical velocities (Δv/a) at the accumulation and ablation stakes on KWF (e.g. the year 2015 refers to the hydrological year 2014/2015). Positive values are defined as emergence flow, negative ones as submergence flow. Location of the stakes s. Figure 1.

Figure 6: Annual horizontal flow velocities (Δs/a) on GPF and TSF and L9 at KWF for comparison (e.g. the year 2015 refers to the hydrological year 2014/2015). GPF (a): Selection of the longitudinal stakes at the tongue of GPF. GPF (b): Three stakes at the cross profile; location s. Figure 1: 71: orogr. left, 73: centre, 75: orogr. right.



Figure 7: Mean daily horizontal velocities (Δs/day) at stake 65 on GPF between the measurements as an example of the interannual fluctuation of surface velocity. The peak in July 2013 shows the highest uncertainty, very likely because of few satellites due to shading effects of the surrounding topography, which depends on the time of the measurements. Additional information is given by the mean elevation change per day from ALS DoDs at the position of the stake (data

5    extended from Stocker-Waldhuber et al. (2017)).



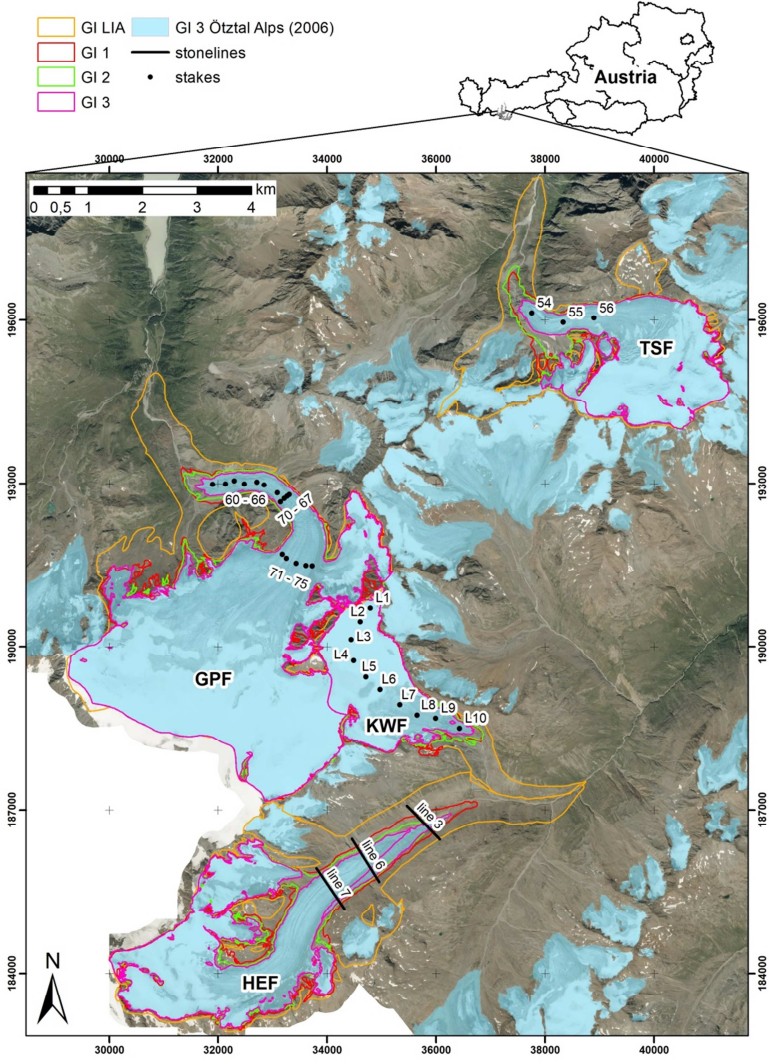

**Figure 1:** Location of the stone lines (3, 6 and 7) on Hintereisferner (HEF) and stakes on Kesselwandferner (KWF), Taschachferner (TSF) and Gepatschferner (GPF). On GPF, stakes 60 to 66 are longitudinal stakes from the glacier snout upwards to the first cross profile 67 to 70 (from the orographic right to the left). Stakes 71 to 75 are located at the root zone of the tongue as a cross profile. The glacier area was taken from the Austrian Glacier Inventories (GI) from LIA (little ice age) around 1850, GI1 from 1969, GI2 from 1998 and GI3 from 2006 (Fischer et al., 2015). Background: Orthophoto from 2015; data source: Land Tirol – data.tirol.gv.at.



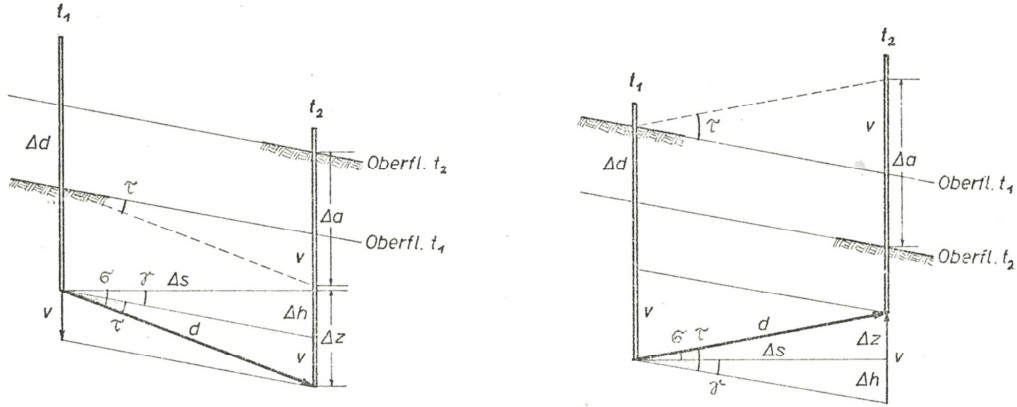

**Figure 2: Drawings by Schneider (1970) of the motion of a stake and changes at the glacier surface (Oberfl.) between two time steps (t1, t2) within the accumulation area (left) and the ablation area (right). d: flow path (length of the velocity vector), v: vertical velocity, Δs: horizontal velocity (projected velocity), Δd: absolute surface elevation change, Δa: relative surface elevation change from accumulation or ablation.**

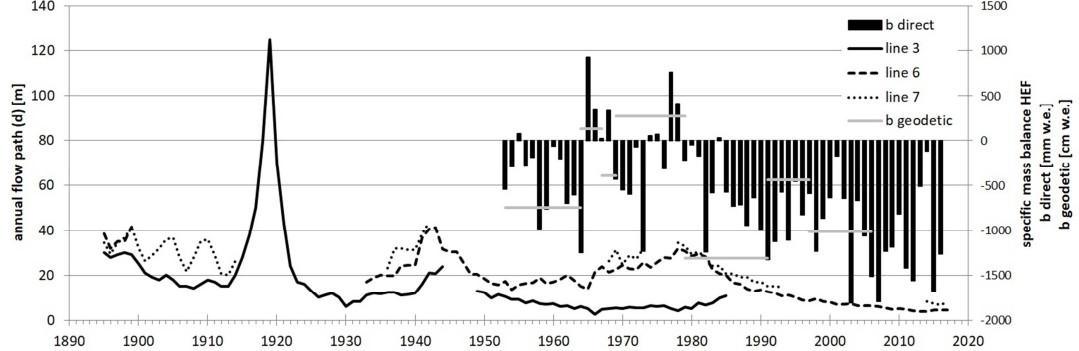

**Figure 3: The mean annual flow path of the stones at Lines 3, 6 and 7 on HEF since 1894/95 (= 1895). Data series extended since Span et al. (1997) and annual specific surface mass balance (b direct) since 1953 (Strasser et al., 2018; WGMS, 2017; original data: Hess, 1924) as well as the geodetic balances from DoDs (b geodetic) by Fischer (2011). Location of the stone lines s. Figure 1.**



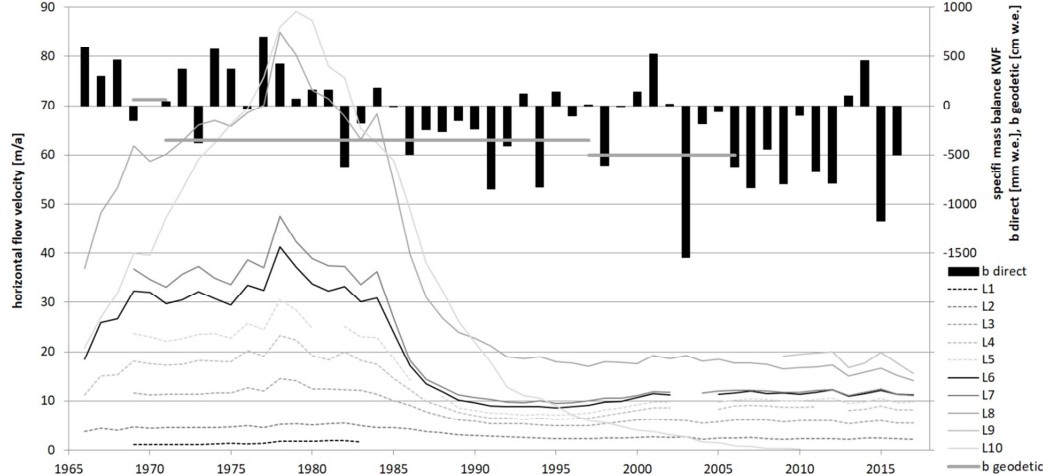

**Figure 4: Annual horizontal flow velocities (Δs/a) at the accumulation and ablation stakes on KWF (e.g. the year 2015 refers to the hydrological year 2014/2015) and the specific surface mass balance (b direct) (Strasser et al., 2018; WGMS, 2017) as well as the geodetic balances from DoDs (b geodetic) by Fischer (2011). Location of the stakes s. Figure 1.**

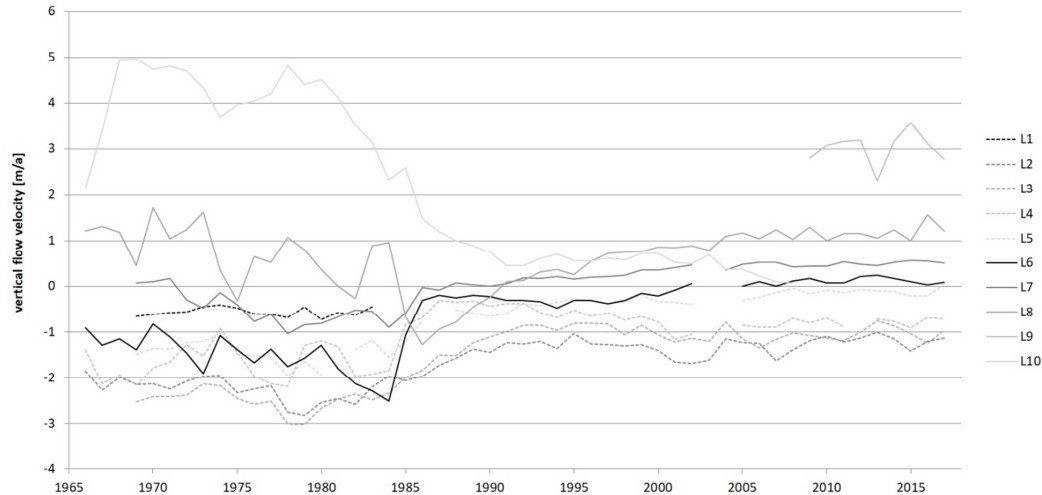

**Figure 5: Annual vertical velocities (Δv/a) at the accumulation and ablation stakes on KWF (e.g. the year 2015 refers to the hydrological year 2014/2015). Positive values are defined as emergence flow, negative ones as submergence flow. Location of the stakes s. Figure 1.**



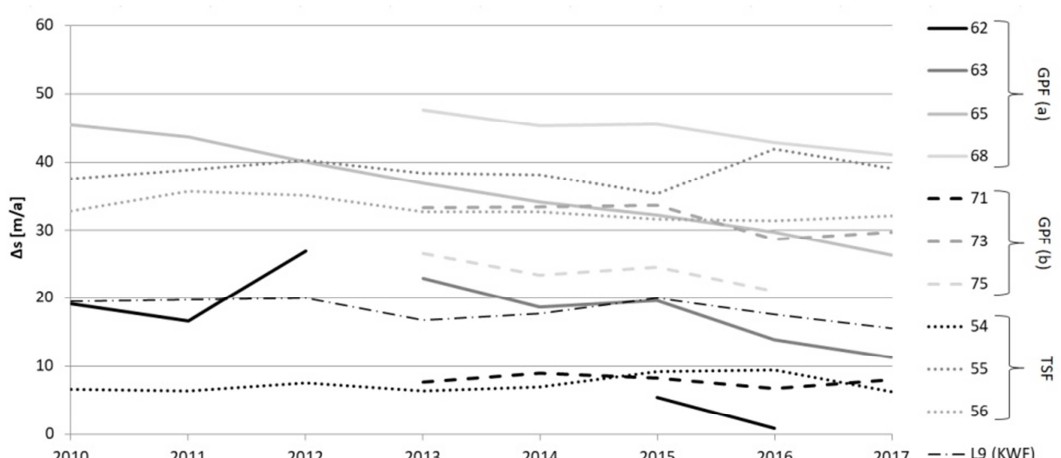

**Figure 6: Annual horizontal flow velocities (Δs/a) on GPF and TSF and L9 at KWF for comparison (e.g. the year 2015 refers to the hydrological year 2014/2015). GPF (a): Selection of the longitudinal stakes at the tongue of GPF. GPF (b): Three stakes at the cross profile; location s. Figure 1: 71: orogr. left, 73: centre, 75: orogr. right.**

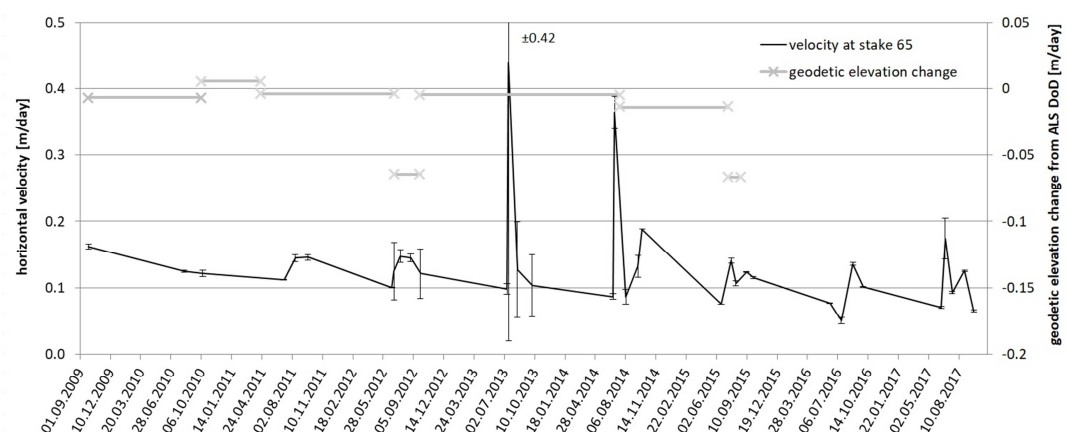

**Figure 7: Mean daily horizontal velocities (Δs/day) at stake 65 on GPF between the measurements as an example of the interannual fluctuation of surface velocity. The peak in July 2013 shows the highest uncertainty, very likely because of few satellites due to shading effects of the surrounding topography, which depends on the time of the measurements. Additional information is given by the mean elevation change per day from ALS DoDs at the position of the stake (data extended from Stocker-Waldhuber et al. (2017)).**