# Peer review of "Ice flow velocity as a sensitive indicator of glacier state"

_The Cryosphere, 2018_

## Referee Comment (RC1) · Anonymous Referee #1 · 6 Apr 2018

Summary: Stocker-Waldhuber et al. presents a vast number of in situ velocity measurements on four Austrian glaciers spanning over one century. They examine the velocity variations in view of changes in mass balances and explore the possibility to use velocity data as an alternative climate indicators.

In principle, such a study would be very welcome. There is a growing number of publications dealing with velocity measurements from satellite data but the interpretation of the velocity variations, often over a period of $\sim$20 years, is far from straightfoward. Accurate field measurements of annual velocity at the same location and during a long period as provided in this study could help to understand the drivers/processes of velocity changes (change in sliding vs. internal deformation) and in turn, facilitate the interpretation of regional velocity variations observed from space. This is what I was

expecting when I read the attractive title "Ice flow velocity as a sensitive indicator of glacier state".

However and although the data are important to present and publish, the paper fails to address (or even explore) this question of the link between velocity change / mass balance / and climate forcing. Many relevant references are missing and the discussion of the results is nearly absent. I am afraid this is currently hardly more than a data paper and I an sorry that I cannot recommend publication in TC.

——

General comments

1/ Seasonal velocity variations are out of the scope of the paper. Poorly presented and not discussed despite an abundant recent literature on the topic.

2/ As said above, the main weakness is the lack of discussion of the results. There is no comparison to other glaciers for which velocity variations have been reported and analyzed. The authors will find at the end of my review a list of publications that, I think, are relevant to their study and hopefully could inspire them to go deeper into their data analysis and discussion. We also miss a clear comparison to mass balance measurements (and a discussion of how/why they are related). Currently this is only shown in Figure 3 and 4.

3/ The title is misleading and results in too much expectation. The reader expects at some point an answer, even partial, to the question "Can large-scale ice flow velocity monitoring be a potential alternative to mass balance measurement for regional glacier monitoring on an annual basis"? This discussion/answer never comes.

——

Specific comments

Abstract. lack of results. Too general.

[Figure]

Introduction. I really enjoyed the historical perspective on glacier monitoring.

——

2.1 I do not really see the point of discussing velocity measurements for the large ice sheets. It is out of the scope of your paper. It seems much more relevant to discuss recent progress (and limitation) of measuring velocity on glaciers. Two references come to my mind (Heid and Kääb, 2012; Dehecq and others, 2015).

2.10 The Jacob et al. paper is from 2012, not 2013. Note that there results, obtained solely from GRACE data are somewhat controversial, especially for regions with a low concentration of glaciers and a high influence of hydrology. (Gardner and others, 2013) is a more accepted global compilation. [see for example the unreliable result for the European Alps]

2.12. I do not understand why the authors take the example of the ELA to illustrate the need to document glacier velocity change. No link. [I do not challenge the value of ELA observations, just that the link with your study is unclear right now]

2.16. Is "parameters" really the right term? "measurements" maybe?

2.17 I find it strange to look for an "alternative". Velocity measurements are valuable by themselves and are eagerly need for improved dynamical modelling of glaciers. No need to replace other variables! They rather complement themselves in understanding the glacier response to climate change.

2.23 why "empirical"? Are not they just "measurements"?

——

3.26. Is not "glaciological" (rather than "direct") the terminology recommended in the UNESCO glossary by Cogley et al.?

3.29 rather "Difference of DEMs" I think ——

4.6 Unclear to me. How do one find in the field the initial (last year) position to make the tape measurements ?

4.9 does it mean that lateral variations (due to lateral drag) are not taken into account? Or are all the stones close to the centerline so that these lateral variations can be neglected?

4.12. this is not a relation but a percentage (or ratio). How variable is this value from year to year? (will depend on the amount of sliding probably). What additional uncertainties arise from this assumption?

4.17 what does " calculated to its base point" means?

4.31 why does "distance" matter for uncertainties in a tape measurement?

4.31 can the authors give us some good reasons to trust their 5% error estimate? Right now we can only believe them.

—-

5.5 "for the period" is vague.

5.19. Can the acceleration in the 1920s (and the 40s) be related to a period of positive mass balance? Is a positive mass balance measured in model such as those of (Marzeion and others, 2012) and (Huss, 2012)? Typically the sort of analysis/discussion that one expect based on these results.

5.22 why is it more obvious in the geodetic results? Do you mean geodetic mass balance? Why not obvious in the glaciological mass balance?

5.24 surface mass balance? Glacier-wide mass balance? Make sure the vocabulary is clear and follow Cogley et al. (2011) glossary. Authors could also provide here and elsewhere % of velocity change to clarify the magnitude of the signal.

—-

6.10 I am not sure the authors defined their convention for vertical motion. Worth reminding in the text anyway. (positive upward I assume)

6.12. Can the authors details/quantify what was this "response". They are generally too vague in the description of their results.

6.16 Can we interpret the lack of velocity change for TSF in term of response time? Has this glacier already reached an equilibrium so that no further velocity change occurs? Would such an interpretation make sense according to your knowledge of this glacier?

6.23. I did not understand the causal link between the depression and a limited increase in velocity

—-

7.8 Can the authors discuss why the response time of the velocity are so different between KWF and HEF?

7.20 Should be named glacier-wide mass balance (even glacier-wide mass balance rate in principle, but I agree that "rate" could be skipped for the sake of simplicity)

7.27 Is this statement that TSF is closer to equilibrium than GPF confirmed in geodetic mass balance using Lidar surveys?

7.28 Do the authors mean seasonal here?

—-

8.3. do the authors mean glacier-wide mass balance? I would say that surface mass balance at selected sites are not influenced by the changes in area (but influenced by changes in elevation through the mass balance-elevation feedback).

—-

Figure 4. check title of the Y-axis. The Y-title use specific, the legend "direct". Make sure terminology is used correctly and follow the UNESCO glossary.

Figure 5 (and4) Do authors need to show all stakes? Why not selecting the ones which are the most informative and used in the text (or show all stakes and highlight some in bold?). Also the velocity seems to be stable for most stakes since 2000. A fact not really discussed in the article.

Good luck for your future work on this important but under-exploited dataset,

——

References about velocity changes on glaciers at annual and decadal timescale (certainly not exhaustive but a good start)

BHATTACHARYA, A., BOLCH, T., MUKHERJEE, K., PIECZONKA, T., KROPÁČEK, J. and BUCHROITHNER, M. F.: Overall recession and mass budget of Gangotri Glacier, Garhwal Himalayas, from 1965 to 2015 using remote sensing data, Journal of Glaciology, 62(236), 1115–1133, doi:10.1017/jog.2016.96, 2016.

Heid, T. and Kääb, A.: Repeat optical satellite images reveal widespread and long term decrease in land-terminating glacier speeds, The Cryosphere, 6, 467–478, doi:10.5194/tc-6-467-2012, 2012.

Mernild, S. H., Knudsen, N. T., Hoffman, M. J., Yde, J. C., Hanna, E., Lipscomb, W. H., Malmros, J. K. and Fausto, R. S.: Volume and velocity changes at Mittivakkat Gletscher, southeast Greenland, Journal of Glaciology, 59(216), 660–670, doi:10.3189/2013JoG13J01, 2013.

Neckel, N., Loibl, D. and Rankl, M.: Recent slowdown and thinning of debris-covered glaciers in south-eastern Tibet, Earth and Planetary Science Letters, 464, 95–102, doi:10.1016/j.epsl.2017.02.008, 2017.

Schaffer, N., Copland, L. and Zdanowicz, C.: Ice velocity changes on Penny Ice Cap, Baffin Island, since the 1950s, Journal of Glaciology, 1–15, doi:10.1017/jog.2017.40, 2017.

Tedstone, A. J., Nienow, P. W., Gourmelen, N., Dehecq, A., Goldberg, D. and Hanna, E.: Decadal slowdown of a land-terminating sector of the Greenland Ice Sheet despite warming, Nature, 526, 692, 2015.

Thomson, L. I. and Copland, L.: Changing contribution of peak velocity events to annual velocities following a multi-decadal slowdown at White Glacier, Annals of Glaciology, 58(75), 145–154, doi:10.1017/aog.2017.46, 2017a.

Thomson, L. I. and Copland, L.: Multi-decadal reduction in glacier velocities and mechanisms driving deceleration at polythermal White Glacier, Arctic Canada, Journal of Glaciology, 63(239), 450–463, doi:10.1017/jog.2017.3, 2017b.

Vincent, C., Soruco, A., Six, D. and Le Meur, E.: Glacier thickening and decay analysis from 50 years of glaciological observations performed on Glacier d'Argentière, Mont Blanc area, France, Annals of Glaciology, 50(50), 73–79, doi:10.3189/172756409787769500, 2009.

Other references cited in the review

Dehecq A, Gourmelen N and Trouve E (2015) Deriving large-scale glacier velocities from a complete satellite archive: Application to the Pamir–Karakoram–Himalaya. Remote Sensing of Environment 162, 55–66 (doi:10.1016/j.rse.2015.01.031)

Gardner AS, Moholdt G, Cogley JG, Wouters B, Arendt AA, Wahr J, Berthier E, Hock R, Pfeffer WT, Kaser G, Ligtenberg SRM, Bolch T, Sharp MJ, Hagen JO, van den Broeke MR and Paul F (2013) A Reconciled Estimate of Glacier Contributions to Sea Level Rise: 2003 to 2009. Science 340(6134), 852–857 (doi:10.1126/science.1234532)

Huss M (2012) Extrapolating glacier mass balance to the mountain-range scale: the European Alps 1900-2100. The Cryosphere 6(4), 713–727 (doi:10.5194/tc-6-713-2012)

Marzeion B, Jarosch AH and Hofer M (2012) Past and future sea-level change from the surface mass balance of glaciers. The Cryosphere 6(6), 1295–1322 (doi:10.5194/tc-6-
1295-2012)
* * *

---

## Author Comment (AC1) · 2 May 2018

We thank the reviewer for the constructive comments, bringing in a new perspective. The comments will help us to improve the manuscript for a broader scientific community, adding additional parts of information which we initially thought was not beneficial to include into the paper. Now we see that it is necessary to provide some background information and a broader context, and look forward to do so.

The underline review is cited underlined, answers of authors are formatted indented.

Review: Summary: Stocker-Waldhuber et al. presents a vast number of in situ velocity measurements on four Austrian glaciers spanning over one century. They examine the velocity variations in view of changes in mass balances and explore the possibility to use velocity data as an alternative climate indicators.

In principle, such a study would be very welcome. There is a growing number of publications dealing with velocity measurements from satellite data but the interpretation of the velocity variations, often over a period of ~20 years, is far from straightforward.

> We fully agree, as this was our motivation to start working on the data.

Accurate field measurements of annual velocity at the same location and during a long period as provided in this study could help to understand the drivers/processes of velocity changes (change in sliding vs. internal deformation) and in turn, facilitate the interpretation of regional velocity variations observed from space. This is what I was expecting when I read the attractive title "Ice flow velocity as a sensitive indicator of glacier state".

> We fully agree, and are glad that the reviewer found the title attractive. The reason for us being very careful in our interpretation of the presented data is that for an in depth understanding governing processes, the ice thickness and temperatures, firn thickness, water content and temperatures, and basal velocities must be known to fully understand velocity variations. For a theoretical or quantitative interpretation of velocity variations, this data must be known not only on the location of surface velocity measurements, but on all other points having an effect on the surface velocity of the measured point. It is, similar to research on atmospheric processes, not possible to determine these parameters at all necessary points, causing equifinality problems at various scales. This is why we decided not to include the topic. We now understand that a presentation of this problem is necessary to set the frame for the analysis carried out. We are lucky to have in situ data of the depth of firn cover at one location at Kesselwandferner (Ambach et al., 1978), so that we are able to illustrate ambiguities resulting from firn thickness variability. We now understand that it could be an important contribution of this paper to not only present the results of ice flow velocity time series, but also the greater glaciological context.

Ambach, W., Blumenthaler, M., Eisner, H., Kirchlechner, P., Schneider, H., Beherens, H., Moser, H., Oerter, H., Rauert, W. und Bergmann, H. (1978): Untersuchungen der Wassertafel am Kesselwandferner (Ötztaler Alpen) an einem 30 Meter tiefen Firnschacht. Zeitschr. f. Gletscherk. und Glazialgeol., 14, 1, p61-71.

However and although the data are important to present and publish, the paper fails to address (or even explore) this question of the link between velocity change / mass balance / and climate forcing. Many relevant references are missing and the discussion of the results is nearly absent. I am afraid this is currently hardly more than a data paper and I an sorry that I cannot recommend publication in TC.

We will include a general introduction on which parameters can influence surface velocity variations, roughly quantify influences of the various parameters (e.g. thickness of firn cover) and go a bit in the depth of discussion. Nevertheless, on one hand we end up in a number of uncertainties, which, on the other hand, have no impact on velocity as empirical monitoring parameter. Thus we will improve the link between velocity change, mass balance and climate forcing, as all the data exist and can be presented straight forward together with the theoretical framework and statistics.

General comments

1/ Seasonal velocity variations are out of the scope of the paper. Poorly presented and not discussed despite an abundant recent literature on the topic.

We measured surface velocities at subseasonal resolution at Gepatschferner and Taschachferner. Subseasonal measurements are not available for the longer time series of Kesselwandferner and Hintereisferner. We will try to improve our presentation and discussion of the data. Currently, velocity data in subseasonal resolution is only presented in Figure 7. We will appreciate to extend this topic. It actually is not clear if the above comment suggests to skip or to extend the topic. For us, both would be ok.

2/ As said above, the main weakness is the lack of discussion of the results. There is no comparison to other glaciers for which velocity variations have been reported and analyzed. The authors will find at the end of my review a list of publications that, I think, are relevant to their study and hopefully could inspire them to go deeper into their data analysis and discussion. We also miss a clear comparison to mass balance measurements (and a discussion of how/why they are related). Currently this is only shown in Figure 3 and 4.

We are very grateful for the hints on additional literature and will work on the comparison of the data as well as on the discussion. We will include a broader discussion of the relation to mass balance and mass balance parameters which we skipped in the original draft to prohibit an 'overload' of the manuscript with the danger of losing focus.

3/ The title is misleading and results in too much expectation. The reader expects at some point an answer, even partial, to the question "Can large-scale ice flow velocity monitoring be a potential alternative to mass balance measurement for regional glacier monitoring on an annual basis"? This discussion/answer never comes.

> We will discuss this topic more explicitly. In the current version, the existing material was not included to keep the original draft focused on ice flow velocities – obviously ending in a too narrow presentation.

Specific comments

Abstract. lack of results. Too general.

> We will add more facts and numbers including additional results from revised parts of the paper as suggested above.

Introduction. I really enjoyed the historical perspective on glacier monitoring.

> Thank you!

2.1 I do not really see the point of discussing velocity measurements for the large ice sheets. It is out of the scope of your paper. It seems much more relevant to discuss recent progress (and limitation) of measuring velocity on glaciers. Two references come to my mind (Heid and Kääb, 2012; Dehecq and others, 2015).

> From the above comments we conclude that a clear focus on mountain glaciers and processes there would be beneficial. The question why in general flow velocity is an interesting parameter seems to be rather obvious, so that this part could be skipped – correct?

2.10 The Jacob et al. paper is from 2012, not 2013. Note that there results, obtained solely from GRACE data are somewhat controversial, especially for regions with a low concentration of glaciers and a high influence of hydrology. (Gardner and others, 2013) is a more accepted global compilation. [see for example the unreliable result for the European Alps]

> With a clear focus on mountain glaciers, this discussion is out of scope. Nevertheless, we thank for the comment.

2.12. I do not understand why the authors take the example of the ELA to illustrate the need to document glacier velocity change. No link. [I do not challenge the value of ELA observations, just that the link with your study is unclear right now]

> The obvious problem that on Alpine glaciers under current climate conditions ELA is undefined/above summits clearly shows the problem of conventional mass balance for present

glacier monitoring. As for surface velocity and its governing parameters, we suggest to add an introduction and additional data plus statistics, improving readability.

2.16. Is "parameters" really the right term? "measurements" maybe?

*"In this paper, four long time series of ice flow velocities are revisited and compared with classical in situ mass balance parameters."*

We want to say that we compare to the results of the measurements, i.e. time series of mass balance parameters. In our opinions, 'measurements' could generally refer to a method, without necessity to refer to the results. We will discuss that with our editorial office.

2.17 I find it strange to look for an "alternative". Velocity measurements are valuable by themselves and are eagerly need for improved dynamical modelling of glaciers. No need to replace other variables! They rather complement themselves in understanding the glacier response to climate change.

In the Alps, conventional mass balance under current climate conditions fails as accumulation often is zero, ELA is above summits and specific mass balance is mainly influenced by loss of ablation area, not the total balance at specific sites. The interpretation of b=B/S for comparing various years fails when S changes more than B.

We agree that there is no necessity to present an alternative, but more a complementary method of establishing time series in glacier monitoring when conventional mass balance method do not provide results any more.

2.23 why "empirical"? Are not they just "measurements"?

As we stated before, we do not generally claim a complete theoretical discussion of our data, and rather stay with an empirical treatment on the relation of velocity data compared to mass balance data/climate forcing. As for example the degree day method can be clearly considered as an empirical method …

It will be completely reworded; we can split these two meanings in using measured data for an empirical analysis.

3.26. Is not "glaciological" (rather than "direct") the terminology recommended in the

UNESCO glossary by Cogley et al.?

We will go over the text to stay close at the Cogley at al. terminology.

3.29 rather "Difference of DEMs" I think

> DEM of Difference is correct (e.g. Wheaton et al. 2010: https://doi.org/10.1002/esp.1886). We can explain that and add the citation.

4.6 Unclear to me. How do one find in the field the initial (last year) position to make the tape measurements?

> The initial position (x,y) is revisited with DGPS and used as starting point for the tape measurements ending at the stone. If the question addresses the fact that the initial z position during recent years was higher than the z position at the time of the second measurement, we can give an error estimate for that. We will discuss that in detail, with a sketchmap.

4.9 does it mean that lateral variations (due to lateral drag) are not taken into account? Or are all the stones close to the centerline so that these lateral variations can be neglected?

> This is an important point - see next comment. We can add some additional information on that topic. – Maybe you could give us some additional hint on the expected depth of the answer (change of slope and elevation at the stone line during years, changing number of stones, profile width?).

4.12. this is not a relation but a percentage (or ratio). How variable is this value from year to year? (will depend on the amount of sliding probably). What additional uncertainties arise from this assumption?

> Thank you for this comment, we will write ratio instead of relation.

> The stones are evenly distributed at the profile, so the lateral variations are taken into account. Thus, the mean value of the stone-velocities is about 80% of the maximum velocity at the centre flow line.

> We will discuss this in more detail. The amount of sliding was measured by comparing the velocity of a stake in the profile with the stone velocity (so far not included in the manuscript).

4.17 what does " calculated to its base point" means?

> The base point is the lowest end of the stake (x,y,z coordinates). We will add this to the text.

4.31 why does "distance" matter for uncertainties in a tape measurement?

> During the measurements the tape touches the ground (at least at a few points) and cannot be stretched perfectly (e.g. in presence of wind). Thus, the surface roughness becomes more important with longer distances which leads to higher uncertainties.

4.31 can the authors give us some good reasons to trust their 5% error estimate? Right now we can only believe them.

We will include the underlying error calculation.

5.5 "for the period" is vague.

Thank you for this comment, we will reword this part. The error is given as the maximum error from two measurements, one in the beginning of the period and the other one at the end. Thus, the measurement accuracy depends on the number of measurements and is independent from the length of the period. We will clarify: ±5 cm per single measurement/survey or at least ±10 cm for the difference between the two readings.

5.19. Can the acceleration in the 1920s (and the 40s) be related to a period of positive mass balance?

Unfortunately, annual mass balance records in Austria started in 1952/53. Length change records show some advances in the 1920. During these years, Kesselwandferner and Hintereisferner separated, so that an interpretation of length change data in terms of mass balance is highly uncertain. Geodetic mass balance data from these periods has low resolution and high uncertainties, so that this hypothesis can be neither confirmed nor rejected. The uncertainties in modelling of past mass balance data of individual glaciers in absence of local high resolution meteorological data might not be allowing to draw a reliable conclusion on the topic of your question. We can add a short not on that.

Fischer, A., G. Patzelt, M. Achrainer, G. Groß, G. K. Lieb, A. Kellerer-Pirklbauer & G. Bendler, 2018. Gletscher im Wandel: 125 Jahre Gletschermessdienst des Alpenvereins. Springer Spektrum, 140 S. doi:10.1007/978-3-662-55540-8. http://www.springer.com/de/book/9783662555392.

Fischer, A., K. Helfricht, H. Wiesenegger, L. Hartl, B. Seiser, M. Stocker-Waldhuber, 2016. Chapter 9 - What Future for Mountain Glaciers? Insights and Implications From Long-Term Monitoring in the Austrian Alps, In: Gregory B. Greenwood and J.F. Shroder, Editor(s), Developments in Earth Surface Processes, Elsevier, 21, 325-382. http://doi.org/10.1016/B978-0-444-63787-1.00009-3

Is a positive mass balance measured in model such as those of (Marzeion and others, 2012) and (Huss, 2012)? Typically the sort of analysis/discussion that one expect based on these results.

Actually we do not think that the focus of the manuscript can include a comparison of model results (including a full discussion of uncertainties) with not-existing in situ mass balance data.

If your question refers to measured mass balance data: yes, the time series include years with positive mass balances.

Measurement data from these periods are very rare (e.g. mass balance on HEF since 1953) and thus, models often fail to reproduce these "early" acceleration rates. This emphasizes the importance of these historical long term data of velocity records. We will discuss this in more detail.

5.22 why is it more obvious in the geodetic results? Do you mean geodetic mass balance? Why not obvious in the glaciological mass balance?

Thank you for the comment. We will delete this sentence and we will add a separate part in the discussion with focus on the difference between the geodetic and the direct glaciological mass balance. E.g. for the period 1969-1979 on HEF in Fig 3 (Fischer, A. (2011): Comparison of direct and geodetic mass balances on a multi-annual time scale. The Cryosphere, 5, 107-124.).

5.24 surface mass balance? Glacier-wide mass balance? Make sure the vocabulary is clear and follow Cogley et al. (2011) glossary. Authors could also provide here and elsewhere % of velocity change to clarify the magnitude of the signal.

We will check the manuscript considering the definitions according to Cogley et al. and we will add some changes on a percentage basis.

6.10 I am not sure the authors defined their convention for vertical motion. Worth reminding in the text anyway. (positive upward I assume)

Thank you for pointing out this gap in our definitions. We define upward motion positive, downward negative, and we will add this to the text (and sketch).

6.12. Can the authors details/quantify what was this "response". They are generally too vague in the description of their results.

Thank you for pointing out the unclear terminology – we will add some state of the art discussion on response and reaction of glaciers to climate change. In addition, we will treat this topic in the revised/enhanced discussion.

6.16 Can we interpret the lack of velocity change for TSF in term of response time? Has this glacier already reached an equilibrium so that no further velocity change occurs? Would such an interpretation make sense according to your knowledge of this glacier?

Thank you for this comment! We will discuss this in more detail in the text, based on thorough definitions of the respective terminology. The lack of velocity change can be explained with the topography of the glacier and the response time. The shorter the glacier tongue will become

the closer the glacier will be to an equilibrium state, but currently the glacier is still far from equilibrium.

**6.23. I did not understand the causal link between the depression and a limited increase in velocity**

„Limited" in this case should mean   limited to the area of the depression. This means during the sink process the velocities (horizontal and vertical) increased within this area. We will clarify this in the text.

**7.8 Can the authors discuss why the response time of the velocity are so different between KWF and HEF?**

We can refer to existing publications why these two glaciers respond different to climate change and some additional data on present state (Kuhn et al., 1985).

Kuhn, M., G. Markl, G. Kaser, U. Nickus, F. Obleitner, H. Schneider, 1985: Fluctuations of climate and mass balance: Different responses of two adjacent glaciers, *Zeitschrift für Gletscherkunde und Glazialgeologie*, 21: 409-416.

**7.20 Should be named glacier-wide mass balance (even glacier-wide mass balance rate in principle, but I agree that "rate" could be skipped for the sake of simplicity)**

*"This is supported by a linear regression of annual mean specific balance (b) of the total glacier area of HEF and KWF versus the mean specific balance of their accumulation areas (bc) for the period 1965/66 – 1999/00 by Span and Kuhn (2003)."*

So do you suggest to change specific mass balance in glacier wide specific mass balance? i.e. not divide by area? Or just rename specific mass balance in specific glacier wide mass balance?

We want to avoid the confusion between specific mass balance and mass balance, and would prefer to use the definition at page 25 of Cogley et al:

> *"Area-averaged (adj.)*
> *Descriptive of a quantity that has been averaged over part or all of the area of the glacier.*
> *The area-averaged mass balance is simply the specific mass balance of the region under*
> *consideration. The adjective has sometimes been used to emphasize that the specific mass*
> *balance is that of the whole glacier and not of a "specific" location (see point mass balance).*
> *"Mean specific mass balance" has been used in the same sense."*

(as defined on page 64)

7.27 Is this statement that TSF is closer to equilibrium than GPF confirmed in geodetic mass balance using Lidar surveys?

Based on a thorough definition of terminology (what means closer to equilibrium?) we will discuss that or cite a discussion of this topic. We will check this in the DoDs and add some comments to the text.

7.28 Do the authors mean seasonal here?

We actually refer to several measurements per year, and will discuss the question with our editorial office. We did not carry out an analysis for accumulation and ablation season.

8.3. do the authors mean glacier-wide mass balance? I would say that surface mass balance at selected sites are not influenced by the changes in area (but influenced by changes in elevation through the mass balance-elevation feedback).

> *"As conventional parameters like ELA tend to be above summit for the investigated glaciers under current conditions and specific mass balance is affected by rapid changes in area […]."*

Specific surface mass balance is defined as glacier wide surface mass balance divided by the area, and therefore clearly influenced by a reduction of area.

Also glacier wide surface mass balance is influenced by the present loss of high ablation zones. We see that is necessary to explain that, and will add some literature. For a first impression how much mass balance changes with area loss see Fischer, 2010.

Fischer, A. (2010) Glaciers and climate change: Interpretation of 50 years of direct mass balance of Hintereisferner, Global and Planetary Change 71, 1-2: 13-26. https://doi.org/10.1016/j.gloplacha.2009.11.014

Figure 4. check title of the Y-axis. The Y-title use specific, the legend "direct". Make sure terminology is used correctly and follow the UNESCO glossary.

It turned out that not title and axis using different labels, rather specific was wrongly used instead of geodetic. We will correct for the revised manuscript.

Figure 5 (and4) Do authors need to show all stakes? Why not selecting the ones which are the most informative and used in the text (or show all stakes and highlight some in bold?). Also the velocity seems to be stable for most stakes since 2000. A fact not really discussed in the article.

We will add a discussion on the "stable" velocities. We aim keeping all stakes to present the overall variability of the velocities along the glacier flow line. We will highlight some of the stakes as suggested.

Good luck for your future work on this important but under-exploited dataset,

Thank you, we will do our best!

References about velocity changes on glaciers at annual and decadal timescale (certainly not exhaustive but a good start)

BHATTACHARYA, A., BOLCH, T., MUKHERJEE, K., PIECZONKA, T., KROPÁˇCEK, J.and BUCHROITHNER, M. F.: Overall recession and mass budget of Gangotri Glacier, Garhwal Himalayas, from 1965 to 2015 using remote sensing data, Journal of Glaciology, 62(236), 1115–1133, doi:10.1017/jog.2016.96, 2016.

Heid, T. and Kääb, A.: Repeat optical satellite images reveal widespread and long term decrease in land-terminating glacier speeds, The Cryosphere, 6, 467–478, doi:10.5194/tc-6-467-2012, 2012.

Mernild, S. H., Knudsen, N. T., Hoffman, M. J., Yde, J. C., Hanna, E., Lipscomb, W. H., Malmros, J. K. and Fausto, R. S.: Volume and velocity changes at Mittivakkat Gletscher, southeast Greenland, Journal of Glaciology, 59(216), 660–670, doi:10.3189/2013JoG13J01, 2013.

Neckel, N., Loibl, D. and Rankl, M.: Recent slowdown and thinning of debris-covered glaciers in south-eastern Tibet, Earth and Planetary Science Letters, 464, 95–102, doi:10.1016/j.epsl.2017.02.008, 2017.

Schaffer, N., Copland, L. and Zdanowicz, C.: Ice velocity changes on Penny Ice Cap, Baffin Island, since the 1950s, Journal of Glaciology, 1–15, doi:10.1017/jog.2017.40, 2017.

Tedstone, A. J., Nienow, P. W., Gourmelen, N., Dehecq, A., Goldberg, D. and Hanna, E.: Decadal slowdown of a land-terminating sector of the Greenland Ice Sheet despite warming, Nature, 526, 692, 2015.

Thomson, L. I. and Copland, L.: Changing contribution of peak velocity events to an nual velocities following a multi-decadal slowdown at White Glacier, Annals of Glaciology, 58(75), 145–154, doi:10.1017/aog.2017.46, 2017a.

Thomson, L. I. and Copland, L.: Multi-decadal reduction in glacier velocities and mechanisms driving deceleration at polythermal White Glacier, Arctic Canada, Journal of Glaciology, 63(239), 450–463, doi:10.1017/jog.2017.3, 2017b.

Vincent, C., Soruco, A., Six, D. and Le Meur, E.: Glacier thickening and decay analysis from 50 years of glaciological observations performed on Glacier d'Argentière, Mont Blanc area, France, Annals of Glaciology, 50(50), 73–79, doi:10.3189/172756409787769500, 2009.

Other references cited in the review

Dehecq A, Gourmelen N and Trouve E (2015) Deriving large-scale glacier velocities from a complete satellite archive: Application to the Pamir–Karakoram–Himalaya. Remote Sensing of Environment 162, 55–66 (doi:10.1016/j.rse.2015.01.031)

Gardner AS, Moholdt G, Cogley JG, Wouters B, Arendt AA, Wahr J, Berthier E, Hock R, Pfeffer WT, Kaser G, Ligtenberg SRM, Bolch T, Sharp MJ, Hagen JO, van den Broeke MR and Paul F (2013) A Reconciled Estimate of Glacier Contributions to Sea Level Rise: 2003 to 2009. Science 340(6134), 852–857 (doi:10.1126/science.1234532)

Huss M (2012) Extrapolating glacier mass balance to the mountain-range scale:  the European  Alps  1900-2100.  The  Cryosphere  6(4),  713–727  (doi:10.5194/tc-6-713-2012)

Marzeion B, Jarosch AH and Hofer M (2012) Past and future sea-level change from the surface mass balance of glaciers. The Cryosphere 6(6), 1295–1322 (doi:10.5194/tc-6-

---

## Referee Comment (RC2) · Anonymous Referee #1 · 3 May 2018

It seems that the authors have more data and ideas that what was included in their initial submission. I then look forward to read a revised version including a deeper discussion of their results and their implications.

Authors asked some clarifications about some of my comments. See below:

Seasonal velocity "It actually is not clear if the above comment suggests to skip or to extend the topic." Right now, the analysis of seasonal velocity variations seems disconnected from the rest of the analysis. I indeed suggested to skip it except if seasonal fluctuations are helpful to understand the multi-year velocity trends.

Ice sheet velocity "The question why in general flow velocity is an interesting parameter seems to be rather obvious, so that this part could be skipped – correct?" The impor-

tance of measuring velocity variations should be explained I think but I suggested to restrict the text to mountain glaciers and remove the ice sheet / ice stream part. (land terminating glaciers around the ice sheets are relevant though)

Jacob et al./Gardner et al. "With a clear focus on mountain glaciers, this discussion is out of scope." Not necessarily. Both papers (Jacob et al., 2012 and Gardner et al., 2013) indeed deal with mountain glaciers (together with ice caps). I just meant that the Gardner et al. study is a more comprehensive assessment because in situ measurements, laser altimetry and Grace data are used and compared.

Repeat measurements of stone position with a tape "The initial position (x,y) is revisited with DGPS and used as starting point for the tape measurements ending at the stone." Given this response, then the next question is: why using a tape and not directly two DGPS positions if you bring a GPS in the field (i.e. measured the location of the stones each year with a DGPS)?

Tranverse Velocity "Maybe you could give us some additional hint on the expected depth of the answer" This is rather the choice of the authors themselves. If they have some data showing velocity variation across the profiles (transverse) I think they are worth reporting (for example the 80% mentioned below could be useful to report). Is the reduction in speed similar all across the profile for example? Is the 80% ratio stable with time? There are not that many tranverse velocity profiles published.

Specific mass balance "So do you suggest to change specific mass balance in glacier wide specific mass balance? i.e. not divide by area? Or just rename specific mass balance in specific glacier wide mass balance? " I was not 100% clear about what "specific" was. So maybe using "average over the accumulation area" is the best option to avoid ambiguity (as Cogley et al., that you quoted in your response, suggested that "specific" has been used in different contexts).

---

## Author Comment (AC2) · 3 May 2018

Thank you for the immediate response and the clarifications!

On seasonal velocity: According to your answer, we will first add all the other additional text, and then see if there is enough space in the revised manuscript to extent the topic and clarify the link to annual velocities. If not, it might be better to skip the seasonal velocities in this manuscript.

Thank you for the clarification for the focus on mountain glaciers and the greater context, we found it hard in the first version to draw a line here. Now we seem to reach a good and solid state of discussion which we are happy with.

Technical question: Why using a tape and not directly two DGPS positions if you bring

a GPS in the field (i.e. measured the location of the stones each year with a DGPS).

This comes from geodetic measurements (theodolite, tachymeter) and early GPS/DGPS measurements, where reliable coordinates have been available only after post processing back in office, and often the accuracy of the positioning was lower than the tape measurements. One of the philosophies in long term monitoring is not to changes techniques more often than necessary, so the tape is still used although during last years (DGPS since 2009) it could have been replaced by DGPS.

Thank you for giving a guideline on variability within the profiles, as well as terminology questions, we will take that as read line for additional information!

---

## Referee Comment (RC3) · M. Pelto (Referee) · 6 May 2018

The authors provide an interesting long term data set on velocity in concert with mass balance on two glaciers Kesselwandferner (KWF) and Hintereisferner (HEF). They provide a short term data set on two others. The data is limited to the ablation zone on HEF. The paper examines the concept that the velocity changes observed might have a comparable use to the annual mass balance or ELA observations. This concept cannot be demonstrated with the spatially poor and temporally infrequent velocity record. The results indicate several long periods of glacier acceleration during periods of positive mass balance, and glacier deceleration during periods of negative mass balance, which has been documented and is expected. The annual mass balance record and terminus change record provide a more detailed record for both glaciers. The terminus

record reflects the cumulative mass balance over a span of years commensurate with a glaciers response time. Velocity is a lagging record to cumulative mass balance over a span of years. This lag varies by glacier and location on the glacier, hence does not provide a good climate record. If the velocity record had a broader spatial distribution on the glacier and had been completed with greater frequency over the period of record, the results would be useful. The level of detail and analysis of HEF and KWF provided by one of the authors Fischer (2011) sets a high bar for mass balance and geodetic analysis of these glaciers that this paper does not add to.

Specific Comments:

1-16: Catastrophic in what sense?

1-23: Remove sentence. No need to discuss ice sheet dynamic changes particularly since these often have retreat associated with acceleration. Ice sheets also have long response times. Marine margins of ice sheets also impacted by ocean temperatures in contact with the ice.

2-3: Not sure the point of this sentence. Feature tracking is quite valuable in velocity determination, particularly to build a rich spatial data set across a glacier and through the seasons. It is true we do not have long term velocity records from feature tracking.

2-12: Not sure how the term mass balance theory is used. The suggestion of velocity replacing mass balance is poorly outlined. Mass balance is a good annual measure of volume change for a glacier. Velocity is a longer term response, not an annual measure it is a lagging indicator (Johannesson et al 1989; Zemp et al. 2015). A mass balance perturbation is distributed over a glacier at a finite rate, which results in a lagged response of both glacier velocity and glacier length to changes in mass balance forcing (Johannesson et al 1989; Bahr et al 1998). Velocity changes can identify changes in state, but these are typically associated with other easily observed changes such as terminus change or snow line rise.

2-14: There have been some years where the ELA has risen above a number of glaciers. A glacier cannot be sustained in such a situation as it has no accumulation zone (Pelto, 2010). The AAR of zero is a good measure in such a year as well and can be identified using satellite imagery as well. Six and Vincent (2014) have illustrated the quantitative relationship between ELA and temperature at a number of glacier indicating the value for understanding climate response. Such a quantitative relationship between climate and velocity has not been established.

5-20: The acceleration is not necessarily the cause of the advance. The increase in mass balance which leads to higher volume flux down glacier would be the formative cause. Schwitter and Raymond (1993) observed some of the complexity noting a difference in longitudinal thickness changes (profile shape factor) for short time-scale changes compared to long time-scale changes illustrating on a short time-scale response where transient responses may cause complex localized thickness-change patterns, as observed here on GPF, and a long time-scale response displaying changes in near steady-state profiles.

5-23: This sentence and figure 3 underscore the insufficiency of the velocity data series as a replacement for annual mass balance. There is a consistent decline in velocity, while this fits the mass balance trend line it does not identify the annual mass balance variability. Berthier and Vincent (2012) in looking at the contribution of mass balance versus surface flux to ice thickness changes on the tongue of Mer de Glace note that "Between 1979–94 and 2000–08, two-thirds of the increase in the thinning rates was caused by reduced ice fluxes and one third by rising surface ablation." To interpret the velocity record for KWF or HEF in the context of climate requires this level of flux analysis is required.

6-9: The pattern of this glacier of acceleration during a period of mass gain followed by reduction during a period of retreat, does not offer any new insights to the glaciers behavior.

6-12: The decline of velocity to nil on KWF raises the same issue as an ELA rising above a glacier it does not offer a specific annual value of use for climate measure.

6-17: The velocities reported for TSF are of insufficient temporal and spatial range to be a useful measure for understanding glacier mass change or climate response.

6-21: The GPF velocity record is of both short duration and generally limited to the terminus reach of the glacier and is not capable of addressing glacier wide velocity changes in response to climate change. Nor can it be demonstrated that velocity change is consistently responsive to mass balance or climate change.

7-5: And what was the response time?

7-10: It is not demonstrated that the simultaneous velocity change is indicative of mass change in a single year. This can be a response due to a similar multi-year mass balance pattern. The multi-year response affect is indicated by the sequence of small positive balances in the early 1980's that led to glacier velocity reduction despite this positive balance, simply because the balance was not as positive as it had been.

8-1: The long term velocity data set presented here is from two glacier HEF and KWF, with data in the accumulation zone only from KWF. This data set does not demonstrate that velocity consistently responds rapidly to climate change. The data indicates decadal velocity responses, not annual velocity changes that are correlated with climate data or mass balance. The velocity observations here do not provide the seasonal details that mass balance records do. This allowed Zemp et al (2015) to observe from the WGMS records a seasonal difference in mass balance "The increased mass loss over the past few decades is driven mainly by summer balances which are dominated in most regions by ablation processes. Winter balances seem to be of secondary importance and show no common trend. As a consequence of both the extended period of mass loss and the delayed dynamic reaction, glaciers in many regions are in strong imbalance with current climatic conditions and, hence, destined to further substantial ice loss." The delayed dynamic response to climate change makes identification of seasonally driven trends difficult. The velocity change smooths out response and will lead to continued velocity change despite short term mass balance changes that could or will influence the glacier.

8-2: The ELA is traditional and may not be the right measure, but the transient snow line elevation on specific dates, including the date when it rises above the glacier if it does, can provide a useful comparable measure (Mernild et al 2013).

Berthier, E., and Vincent, C.: Relative contribution of surface mass-balance and ice-flux changes to the accelerated thinning of Mer de Glace, French Alps, over 1979-2008. Journal of Glaciology, 58(209), 501-512. doi:10.3189/2012JoG11J083, 2012.

Fischer, A.: Comparison of direct and geodetic mass balances on a multi-annual time scale, The Cryosphere, 5, 107-124, https://doi.org/10.5194/tc-5-107-2011, 2011.

Johannesson, T., Raymond, C., and Waddington E.: Time-scale for adjustment of glacier to changes in mass balance, J. Glaciol.,35(121), 355–369, 1989.

Mernild, S., Pelto, M., Malmros, J., ; Yde, J., Knudsen, N., ; Hanna, E.:: Identification of snow ablation rate, ELA, AAR and net mass balance using transient snowline variations on two Arctic glaciers. Journal of Glaciology, 59, 649-659, 2013.

Pelto, M.: Forecasting temperate alpine glacier survival from accumulation zone observations. The Cryosphere, 4, 67–75, 2010.

Schwitter, M. P. and Raymond, C.: Changes in the longitudinal profile of glaciers during advance and retreat, J. Glaciol., 39(133), 582–590, 1993.

Six, D., & Vincent, C.: Sensitivity of mass balance and equilibrium-line altitude to climate change in the French Alps. Journal of Glaciology, 60(223), 867-878. doi:10.3189/2014JoG14J014, 2014.

Zemp, M., and 16 others. Historically unprecedented global glacier decline in the early 21st century. J. Glaciol. 2015, 61 (228), 745-762. doi: 10.3189/2015JoG15J017, 2015.

---

## Author Comment (AC3) · 18 May 2018

The review is cited underlined, answers of authors are formatted indented.

> We thank Mauri Pelto for his helpful suggestions and valuable discussion of the draft! His review, together with the suggestion of the first reviewer, points out that we should clarify the research question, the framework and setting and intensify the discussion of the results. We look forward to do so, following the suggestions of both reviewers.

The authors provide an interesting long term data set on velocity in concert with mass balance on two glaciers Kesselwandferner (KWF) and Hintereisferner (HEF). They provide a short term data set on two others. The data is limited to the ablation zone on HEF.

> Thank you for the interest in the long term data set. For further discussion: it is not entirely clear to us which of the presented data you refer to long term: Is it the Hintereisferner series only (ablation zone), or also the Kesselwandferner series (ablation and accumulation zone)? As these do not include seasonal, but only annual data, we thought it would be useful to add the shorter data sets with higher temporal resolution.

The paper examines the concept that the velocity changes observed might have a comparable use to the annual mass balance or ELA observations. This concept cannot be demonstrated with the spatially poor and temporally infrequent velocity record. The results indicate several long periods of glacier acceleration during periods of positive mass balance, and glacier deceleration during periods of negative mass balance, which has been documented and is expected.

> It is correct that all available velocity data measured in field or by remote sensing, are discontinuous in space and time, and, for remote sensing and stone line measurements, are limited to two dimensions only introducing uncertainties. The authors compared measured mass balance data only, so that they did not draw the conclusion that during the period of rising velocities in the 1920s with rare documentation of glacier advances also positive mass balances did occur. There is only one period of measured positive mass balances (at Kesselwandferner only), where also velocity increases have been measured. As for the first review of this draft, the authors find these comments helpful to work out a better focus of the draft on more explicit research questions. There is enough (maybe too much material) available, so that we will do our best to focus and structure the paper in a new way.

The annual mass balance record and terminus change record provide a more detailed record for both glaciers. The terminus record reflects the cumulative mass balance over a span of years commensurate with a glaciers response time. Velocity is a lagging record to cumulative mass balance over a span of years. This lag varies by glacier and location on the glacier, hence does not provide a good climate record. If the velocity record had a broader spatial distribution on the glacier and had been completed with greater frequency over the period of record, the results would be useful. The

level of detail and analysis of HEF and KWF provided by one of the authors Fischer (2011) sets a high bar for mass balance and geodetic analysis of these glaciers that this paper does not add to.

We will restructure the paper using the comments of Mauri Pelto as guidelines. The phrasing in the first draft seems to very misleading: Our intention was not primarily to present velocity records as an alternative to mass balance observations, but rather contributing complementary information to direct or geodetic mass balance records or other parameters as snow line, etc. This could be useful when ELA rises above summit level and, in a historical perspective, when direct mass balance measurements are incomplete (e.g. before 1953 on HEF).

It seems that the title of the manuscript is misleading so the expectations of the reader are not fulfilled. We will change the title.

We will restructure the whole manuscript in two main parts or even two paper drafts. In the first part we will present the valuable data useful for e.g. glacier modelling. In the second part we will extend the discussion and add time series of temperature and precipitation from the HISTALP data set (e.g. Auer et al. 2007): http://www.zamg.ac.at/histalp/

Specific Comments:

1-16: Catastrophic in what sense?

Because of the glacier advances, the river Rofenache was dammed several times during the little ice age. The failure of the dam caused catastrophic lake outburst floods, i.e causing large damages and devastating the inhabited valley. We will add this information and citations.

1-23: Remove sentence. No need to discuss ice sheet dynamic changes particularly since these often have retreat associated with acceleration. Ice sheets also have long response times. Marine margins of ice sheets also impacted by ocean temperatures in contact with the ice.

We will remove this sentence.

2-3: Not sure the point of this sentence. Feature tracking is quite valuable in velocity determination, particularly to build a rich spatial data set across a glacier and through the seasons. It is true we do not have long term velocity records from feature tracking.

Yes, we actually consider feature tracking or DINSAR as very important methods for monitoring glaciers (and ice caps, ice sheets…). Having worked with these methods at small mountain glaciers, we just wondered about the relationship between the results of the various methods including in situ data. So far, few comparisons of remote sensing and in situ data exist. The interpretation of the direction of motion from remote sensing usually lacks one dimension. We

will discuss this in more detail, although this topic could be enough material for a draft itself, adding remote sensing data of the area.

2-12: Not sure how the term mass balance theory is used. The suggestion of velocity replacing mass balance is poorly outlined. Mass balance is a good annual measure of volume change for a glacier. Velocity is a longer term response, not an annual measure it is a lagging indicator (Johannesson et al 1989; Zemp et al. 2015). A mass balance perturbation is distributed over a glacier at a finite rate, which results in a lagged response of both glacier velocity and glacier length to changes in mass balance forcing (Johannesson et al 1989; Bahr et al 1998). Velocity changes can identify changes in state, but these are typically associated with other easily observed changes such as terminus change or snow line rise.

We will add an introduction and some citation what we refer to as mass balance theory. In short, it is the theoretical framework allowing the interpretation of a measured value based on general theory of science. Apart from teaching books, the cited papers together with the books of Oerlemans should give a good basis. We will try to quantify the mass balance/velocity response time lag!

We do not want to replace direct mass balance measurements, but we want to show if or how velocity records can complement these datasets. We agree that the response time/time lag is a major issue. We will clarify this and discuss this in more detail.

2-14: There have been some years where the ELA has risen above a number of glaciers. A glacier cannot be sustained in such a situation as it has no accumulation zone (Pelto, 2010). The AAR of zero is a good measure in such a year as well and can be identified using satellite imagery as well. Six and Vincent (2014) have illustrated the quantitative relationship between ELA and temperature at a number of glacier indicating the value for understanding climate response. Such a quantitative relationship between climate and velocity has not been established.

From the results of the last decade of direct mass balance measurements, we agree that a repeat AAR of zero is indicating a potentially vanishing glacier. Nevertheless, melt rates can differ significantly, depending not only on the time when the AAR occurred for the first time in the season, so that we consider additional information as helpful. As the time and the rate for a glacier to vanish might be interesting at least for local stake holders, we think that it might be helpful to extend monitoring for these years even in absence of many stakes (which are often hard to maintain on disintegrating glaciers with much debris cover.

As some of the data describing glacier retreat in the Autrian Alps point out the possibility of nonlinear behaviour (e.g. Stocker-Waldhuber et al., 2016), we cannot confirm that the relationships between temperature and mass balance also presented e.g. by Kuhn for the Austrian Alps will also be valid in near future. Nevertheless, we are grateful for pointing out that it will be beneficial to add some time series of temperatures and precipitation from the

HISTALP dataset (Auer et al. 2007) and discuss the relationship to various glaciological parameters in the paper. The disintegration of large glaciers into small ones in higher elevation causes a higher frequency of missing ELAs: Glaciers are either totally snow covered, or no accumulation takes place.

5-20: The acceleration is not necessarily the cause of the advance. The increase in mass balance which leads to higher volume flux down glacier would be the formative cause. Schwitter and Raymond (1993) observed some of the complexity noting a difference in longitudinal thickness changes (profile shape factor) for short time-scale changes compared to long time-scale changes illustrating on a short time-scale response where transient responses may cause complex localized thickness-change patterns, as observed here on GPF, and a long time-scale response displaying changes in near steady-state profiles.

Thank you, we will add this information to the text. For the first draft, we ignored thickness and density changes as well as basal sliding completely, a strategy which we recognized is not useful for a further discussion ….

5-23: This sentence and figure 3 underscore the insufficiency of the velocity data series as a replacement for annual mass balance. There is a consistent decline in velocity, while this fits the mass balance trend line it does not identify the annual mass balance variability. Berthier and Vincent (2012) in looking at the contribution of mass balance versus surface flux to ice thickness changes on the tongue of Mer de Glace note that "Between 1979–94 and 2000–08, two-thirds of the increase in the thinning rates was caused by reduced ice fluxes and one third by rising surface ablation." To interpret the velocity record for KWF or HEF in the context of climate requires this level of flux analysis is required.

We carried out flux analyses as part of a very different paper. Unless velocities are very low, the unknown basal sliding makes a large difference. But that can be shown straight forward in an introduction.

We will clarify in the text that we do not want to replace mass balance observations.

6-9: The pattern of this glacier of acceleration during a period of mass gain followed by reduction during a period of retreat, does not offer any new insights to the glaciers behavior.

We see that we significantly will have to improve the description of the data, pointing out the remarkable events.

6-12: The decline of velocity to nil on KWF raises the same issue as an ELA rising above a glacier it does not offer a specific annual value of use for climate measure.

In contrast to ELA, the measurements did not decline to nil at every stake at KWF, sorry for the misleading phrasing here. We will point out that the 'inactive' fraction of a glacier itself can provide some information.

6-17: The velocities reported for TSF are of insufficient temporal and spatial range to be a useful measure for understanding glacier mass change or climate response.

We will show the differences between the individual glaciers and discuss this in more detail. The added value of the velocity measurements on GPF and TSF is the relatively high velocity for recent glacier states. On GPF and TSF we like to show also the seasonal variability.

6-21: The GPF velocity record is of both short duration and generally limited to the terminus reach of the glacier and is not capable of addressing glacier wide velocity changes in response to climate change. Nor can it be demonstrated that velocity change is consistently responsive to mass balance or climate change.

We will discuss this in more detail as we want to show the seasonal variability, we will also add some values of the surface ablation at the stake positions. The measurements are limited to the glacier tongue but the velocity records in this region can provide information on the glacier state.

7-5: And what was the response time?

Great question. We see that we need to define, discuss and quantify response time, lag, relaxation, delay …

7-10: It is not demonstrated that the simultaneous velocity change is indicative of mass change in a single year. This can be a response due to a similar multi-year mass balance pattern. The multi-year response affect is indicated by the sequence of small positive balances in the early 1980's that led to glacier velocity reduction despite this positive balance, simply because the balance was not as positive as it had been.

We thank for the valuable suggestion and see that we have to present a more elaborated and distinct analysis of the data.

8-1: The long term velocity data set presented here is from two glacier HEF and KWF, with data in the accumulation zone only from KWF. This data set does not demonstrate that velocity consistently responds rapidly to climate change. The data indicates decadal velocity responses, not annual velocity changes that are correlated with climate data or mass balance.

We will carry out a statistical analysis to confirm or decline this hypothesis – thank you for suggesting this hypothesis

The velocity observations here do not provide the seasonal details that mass balance records do.

Unfortunately, seasonal mass balances are not available for the measured glaciers (no winter mass balance is available).

This allowed Zemp et al (2015) to observe from the WGMS records a seasonal difference in mass balance "The increased mass loss over the past few decades is driven mainly by summer balances which are dominated in most regions by ablation processes. Winter balances seem to be of secondary importance and show no common trend. As a consequence of both the extended period of mass loss and the delayed dynamic reaction, glaciers in many regions are in strong imbalance

Another important point of discussion is the definition and quantification of imbalance…

with current climatic conditions and, hence, destined to further substantial ice loss." The delayed dynamic response to climate change makes identification of seasonally driven trends difficult. The velocity change smooths out response and will lead to continued velocity change despite short term mass balance changes that could or will influence the glacier.

We will check that in the data.

8-2: The ELA is traditional and may not be the right measure, but the transient snow line elevation on specific dates, including the date when it rises above the glacier if it does, can provide a useful comparable measure (Mernild et al 2013).

This is a great possibility which provides important information in such years, but there is a lack of information on these changes in the sense of long term investigations. We will add observations of the transient snow line and the citation here.

Berthier, E., and Vincent, C.: Relative contribution of surface mass-balance and ice-flux changes to the accelerated thinning of Mer de Glace, French Alps, over 1979-2008.

Journal of Glaciology, 58(209), 501-512. doi:10.3189/2012JoG11J083, 2012.

Fischer, A.: Comparison of direct and geodetic mass balances on a multi-annual time scale, The Cryosphere, 5, 107-124, https://doi.org/10.5194/tc-5-107-2011, 2011.

Johannesson, T., Raymond, C., and Waddington E.: Time-scale for adjustment of glacier to changes in mass balance, J. Glaciol.,35(121), 355–369, 1989.

Mernild, S., Pelto, M., Malmros, J., ; Yde, J., Knudsen, N., ; Hanna, E.:: Identification of snow ablation rate, ELA, AAR and net mass balance using transient snowline variations on two Arctic glaciers. Journal of Glaciology, 59, 649-659, 2013.

Pelto, M.: Forecasting temperate alpine glacier survival from accumulation zone observations. The Cryosphere, 4, 67–75, 2010.

Schwitter, M. P. and Raymond, C.: Changes in the longitudinal profile of glaciers during advance and retreat, J. Glaciol., 39(133), 582–590, 1993.

Six, D., & Vincent, C.: Sensitivity of mass balance and equilibrium-line altitude to climate change in the French Alps. Journal of Glaciology, 60(223), 867-878.

doi:10.3189/2014JoG14J014, 2014.

Zemp, M., and 16 others. Historically unprecedented global glacier decline in the early 21st century. J. Glaciol. 2015, 61 (228), 745-762. doi: 10.3189/2015JoG15J017, 2015.

---

## Editor Comment (EC1) · G. Catania (Editor) · 21 May 2018

Based on the comments from both reviewers and my view as associate editor, I am rejecting this manuscript. In large part this is because the reviewer comments are quite extensive and critical indicating that the manuscript does not deliver on the proposed topic. I also anticipate that any revised manuscript will depart significantly from the original submission in several fundamental ways. Thus, I recommend that the manuscript is rejected in its current state.

The reviewers have provided numerous helpful suggestions that will aid preparation of a future resubmission of a different manuscript. Importantly, the manuscript would benefit from a re-focus away from an explicit exploration of ice flow velocity as an

indicator of mass balance (as the title indicates) because it is not feasible when velocity changes lag mass balance changes by a few years (as M. Pelto suggests). Instead, the reviewers both commended the value of long-term observations of glacier velocity and suggest going into more details on these data with comparison to other studies for which velocity variations have been reported. Both reviewers provide numerous suggested publications to explore for this work and both are enthusiastic about the value of long-term velocity observations. Thus, I anticipate a revised manuscript that takes into account these extensive reviewer comments will be a welcome submission to TC in the future.

---

## Author Comment (AC4) · 24 May 2018

We thank the editor and the two referees for their helpful suggestions and their valuable discussion on the manuscript. We agree to the decision of the editor.

In a first step we will prepare a data-paper on the basis of this draft by considering the comments of the discussion. This manuscript will be presented under a new title which clearly focuses on the long-term datasets of glacier surface velocities. We think that this data-paper than should be published in a specific data journal e.g. in ESSD.

In a second step we will submit a new manuscript with the existing title by following the suggestions of the referees from the discussion. This manuscript will focus on further analyses of the data especially regarding response times, temperature and

precipitation from HISTALP data-sets, together with direct and geodetic mass balances and glacier length change records. We suppose that this new manuscript will fulfill the expectations arising from the title.